# Approximation capability of neural networks on spaces of probability measures and tree-structured domains

## Abstract

This paper extends the proof of density of neural networks in the space of continuous (or even measurable) functions on Euclidean spaces to functions on compact sets of probability measures. By doing so the work parallels a more then a decade old results on mean-map embedding of probability measures in reproducing kernel Hilbert spaces. The work has wide practical consequences for multi-instance learning, where it theoretically justifies some recently proposed constructions. The result is then extended to Cartesian products, yielding universal approximation theorem for tree-structured domains, which naturally occur in data-exchange formats like JSON, XML, YAML, AVRO, and ProtoBuffer. This has important practical implications, as it enables to automatically create an architecture of neural networks for processing structured data (AutoML paradigms), as demonstrated by an accompanied library for JSON format.

## 1 Motivation

Prevalent machine learning methods assume their input to be a vector or a matrix of a fixed dimension, or a sequence, but many sources of data have the structure of a tree, imposed by data formats like JSON, XML, YAML, Avro, or ProtoBuffer (see Figure 1 for an example). While the obvious complication is that such a tree structure is more complicated than having a single variable, these formats also contain some "elementary" entries which are already difficult to handle in isolation. Beside strings, for which a plethora conversions to real-valued vectors exists (one-hot encoding, histograms of n-gram models, word2vec (Mikolov et al., 2013), output of a recurrent network, etc.), the most problematic elements seem to be unordered lists (sets) of records (such as the "workouts" element and all of the subkeys of "speedData" in Figure 1), whose length can differ from sample to sample and the classifier processing this input needs to be able to cope with this variability.

```
{"weekNumber":"39",
"workouts":[
{ "sport":"running",
  "distance":19738,
  "duration":1500,
  "calories":375,
  "avgPace":76,
  "speedData":{
    "speed":[10,9,8],
    "altitude":[100,104,103,81],
    "labels":["0.0km","6.6km
        ","13.2km","19.7km"]}},
{"sport":"swimming",
  "distance":664,
  "duration":1800,
  "calories":250,
  "avgPace":2711}]}
```

Figure 1: Example of JSON document, adapted from https://github.com/vaadin/fitness-tracker-demo

The variability exemplified above by "workouts" and "speedData" is the defining feature of *Multi-instance learning* (MIL) problems (also called *Deep Sets* in Zaheer et al. (2017)), where it is intuitive to define a sample as a collection of feature vectors. Although all vectors within the collection have the same dimension, their number can differ from sample to sample. In MIL nomenclature, a sample is called a *bag* and an individual vector an *instance*. The difference between *sequences* and bags is that the order of instances in the bag is not important and the output of the classifier should be the same for an arbitrary permutation of instances in the vector.

MIL was introduced in Dietterich et al. (1997) as a solution for a problem of learning a classifier on instances from labels available on the level of a whole bag. To date, many approaches to solve the

problem have been proposed, and the reader is referred to Amores (2013) for an excellent review and taxonomy. The setting has emerged from the assumption of a bag being considered positive if at least one instance was positive. This assumption is nowadays used for problems with weakly-labeled data (Bergamo & Torresani, 2010). While many different definitions of the problem have been introduced (see Foulds & Frank (2010) for a review), this work adopts a general definition of Muandet et al. (2012), where each sample (bag) is viewed as a probability distribution observed through a set of realizations (instances) of a random variable with this distribution. Rather than working with vectors, matrices or sequences, the classifier therefore classifies *probability measures*.

Independent works of Zaheer et al. (2017); Edwards & Storkey (2017) and Pevný & Somol (2017) have proposed an adaptation of neural networks to MIL problems (hereinafter called MIL NN). The adaptation uses two feed-forward neural networks, where the first network takes as an input individual instances, its output is an element-wise averaged, and the resulting vector describing the whole bag is sent to the second network. This simple approach yields a very general, well performing and robust algorithm, which has been reported by all three works. Since then, the MIL NN has been used in numerous applications, for example in causal reasoning (Santoro et al., 2017), in computer vision to process point clouds (Su et al., 2018; Xu et al., 2018), in medicine to predict prostate cancer (Ing et al., 2018), in training generative adversarial networks (Ing et al., 2018), or to process network traffic to detect infected computers (Pevný & Somol, 2016). The last work has demonstrated that the MIL NN construction can be nested (using sets of sets as an input), which allows the neural network to handle data with a *hierarchical* structure.

The wide-spread use of neural networks is theoretically justified by their universal approximation property – the fact that any continuous function on (a compact subset of) a Euclidean space to real numbers can be approximated by a neural network with arbitrary precision (Hornik, 1991; Leshno et al., 1993). However, despite their good performance and increasing popularity, no general analogy of the universal approximation theorem has been proven for MIL NNs. This would require showing that MIL NNs are dense in the space of continuous functions from the space of probability measures to real numbers and – to the best of our knowledge – the only result in this direction is restricted to input domains with finite cardinality (Zaheer et al., 2017).

This work fills this gap by formally proving that MIL NNs with two non-linear layers, a linear output layer and mean aggregation after the first layer are dense in the space of continuous functions from the space of probability measures to real numbers (Theorem 2 and Corollary 3). In Theorem 5, the proof is extended to data with an arbitrary tree-like schema (XML, JSON, ProtoBuffer). The reasoning behind the proofs comes from kernel embedding of distributions (mean map) (Smola et al., 2007; Sriperumbudur et al., 2008) and related work on Maximum Mean Discrepancy (Gretton et al., 2012). This work can therefore be viewed as a formal adaptation of these tools to neural networks. While these results are not surprising, the authors believe that as the number of applications of NNs to MIL and tree-structured data grows, it becomes important to have a formal proof of the soundness of this approach.

The paper only contains theoretical results — for experimental comparison to prior art, the reader is referred to Zaheer et al. (2017); Edwards & Storkey (2017); Pevný & Somol (2017); Santoro et al. (2017); Su et al. (2018); Xu et al. (2018); Ing et al. (2018); Pevný & Somol (2016). However, the authors provide a proof of concept demonstration of processing JSON data at `https://codeocean.com/capsule/182df525-8417-441f-80ef-4d3c02fea970/?ID=f4d3be809b14466c87c45dfabbaccd32`.

## 2 NOTATION AND SUMMARY OF RELEVANT WORK

This section provides background for the proposed extensions of the universal approximation theorem (Hornik, 1991; Leshno et al., 1993). For convenience, it also summarizes solutions to multi-instance learning problems proposed in Pevný & Somol (2017); Edwards & Storkey (2017).

By $C(\mathcal{K}, \mathbb{R})$ we denote the space of continuous functions from $\mathcal{K}$ to $\mathbb{R}$ endowed with the topology of uniform convergence. Recall that this topology is metrizable by the supremum metric $||f - g||_{\sup} = \sup_{x \in K} |f(x) - g(x)|$.

Throughout the text, $\mathcal{X}$ will be an arbitrary metric space and $\mathcal{P}_{\mathcal{X}}$ will be some compact set of (Borel) probability measures on $\mathcal{X}$. Perhaps the most useful example of this setting is when $\mathcal{X}$

is a compact metric space and $\mathcal{P}_{\mathcal{X}} = \mathcal{P}(\mathcal{X})$ is the space of *all* Borel probability measures on $\mathcal{X}$. Endowing $\mathcal{P}_{\mathcal{X}}$ with the $w^{\star}$ topology turns it into a compact metric space (the metric being $\rho^{*}(p, q) = \sum_{n} 2^{-n} \cdot |\int f_{n}\mathrm{d}p - \int f_{n}\mathrm{d}q|$ for some dense subset $\{f_{n} \,|\, n \in \mathbb{N}\} \subset C(\mathcal{X}, \mathbb{R})$ – see for example Proposition 62 from Habala et al. (1996)). Alternatively, one can define metric on $\mathcal{P}(\mathcal{X})$ using for example integral probability metrics (Müller, 1997) or total variation. In this sense, the results presented below are general, as they are not tied to any particular topology.

## 2.1 Universal approximation theorem on compact subsets of $\mathbb{R}^d$

The next definition introduces set of affine functions forming the base of linear and non-linear layers of neural networks.

**Definition 1.** *For any $d \in \mathbb{N}$, $\mathcal{A}^d$ is the set of all affine functions on $\mathbb{R}^d$ i.e.*

$$\mathcal{A}^d = \left\{a : \mathbb{R}^d \to \mathbb{R} \,|\, a(x) = w^{\mathrm{T}}x + b, w \in \mathbb{R}^d, b \in \mathbb{R}\right\}. \tag{1}$$

The main result of Leshno et al. (1993) states that feed-forward neural networks with a single non-linear hidden layer and linear output layer (hereinafter called $\Sigma$-networks) are dense in the space of continuous functions. Lemma 1.1 then implies that the same holds for measurable functions.

**Theorem 1** (Universal approximation theorem on $\mathbb{R}^d$). *For any non-polynomial measurable function $\sigma$ on $\mathbb{R}$ and every $d \in \mathbb{N}$, the following family of functions is dense in $C(\mathbb{R}^d, \mathbb{R})$:*

$$\Sigma(\sigma, \mathcal{A}^d) = \left\{f : \mathbb{R}^d \to \mathbb{R} \,\middle|\, f(x) = \sum_{i=1}^{n} \alpha_i \sigma(a_i(x)), n \in \mathbb{N}, \alpha_i \in \mathbb{R}, a_i \in \mathcal{A}^d\right\}. \tag{2}$$

The key insight of the theorem isn't that a single non-linear layer suffices, but the fact that any continuous function can be approximated by neural networks. Recall that for $\mathcal{K} \subset \mathbb{R}^d$ compact, any $f \in C(\mathcal{K}, \mathbb{R})$ can be continuolusly extended to $\mathbb{R}^d$, and thus the same result holds for $C(\mathcal{K}, \mathbb{R})$. Note that if $\sigma$ was a polynomial of order $k$, $\Sigma(\sigma, \mathcal{A}^d)$ would only contain polynomials of order $\leq k$.

The following metric corresponds to the notion of convergence in measure:

**Definition 2** (Def. 2.9 from Hornik (1991)). *For a Borel probability measure $\mu$ on $\mathcal{X}$, define a metric*

$$\rho_{\mu}(f, g) = \inf \left\{\epsilon > 0 \,|\, \mu\left(\{x \in \mathcal{X}; |f(x) - g(x)| \geq \epsilon\}\right) < \epsilon \right\} \tag{3}$$

*on $M(\mathcal{X}, \mathbb{R})$, where $M(\mathcal{X}, \mathbb{R})$ denotes the collection of all (Borel) measurable functions.*

Note that for finite $\mu$, the uniform convergence implies convergence in $\rho_{\mu}$ (Hornik, 1991, L. A.1):

**Lemma 1.1.** *For every finite Borel measure $\mu$ on a compact $\mathcal{K}$, $C(\mathcal{K}, \mathbb{R})$ is $\rho_{\mu}$-dense in $M(\mathcal{K}, \mathbb{R})$.*

## 2.2 Multi-instance neural networks

In Multi-instance learning it is assumed that a sample $\mathbf{x}$ consists of multiple vectors of a fixed dimension, i.e. $\mathbf{x} = \{x_1, \ldots, x_l\}$, $x_i \in \mathbb{R}^d$. Furthermore, it is assumed that labels are provided on the level of samples $\mathbf{x}$, rather than on the level of individual instances $x_i$.

To adapt feed-forward neural networks to MIL problems, the following construction has been proposed in Pevný & Somol (2017); Edwards & Storkey (2017). Assuming mean aggregation function, the network consists of two feed-forward neural networks $\phi : \mathbb{R}^d \to \mathbb{R}^k$ and $\psi : \mathbb{R}^k \to \mathbb{R}^o$. The output of function is calculated as follows:

$$f(\mathbf{x}) = \psi\left(\frac{1}{l}\sum_{i=1}^{l} \phi(x_i)\right), \tag{4}$$

where $d$, $k$, $o$ is the dimension of the input, output of the first neural network, and the output. This construction also allows the use of other aggregation functions such as maximum.

The general definition of a MIL problem (Muandet et al., 2012) adopted here views instances $x_i$ of a single sample $\mathbf{x}$ as realizations of a random variable with distribution $p \in \mathcal{P}_{\mathcal{X}}$, where $\mathcal{P}_{\mathcal{X}}$ is a set of probability measures on $\mathcal{X}$. This means that the sample is not a single vector but a probability distribution observed through a finite number of realizations of the corresponding random variable.

The main result of Section 3 is that the set of neural networks with (i) $\phi$ being a single non-linear layer, (ii) $\psi$ being one non-linear layer followed by a linear layer, and (iii) the aggregation function being mean as in Equation (4) is dense in the space $C(\mathcal{P}_\mathcal{X}, \mathbb{R})$ of continuous functions on any compact set of probability measures. Lemma 1.1 extends the result to the space of measurable functions.

The theoretical analysis assumes functions $f : \mathcal{P}_\mathcal{X} \to \mathbb{R}$ of the form

$$f(p) = \psi \left( \int \phi(x) \mathrm{d}p(x) \right), \tag{5}$$

whereas in practice $p$ can only be observed through a finite set of observations $\mathbf{x} = \{x_i \sim p | i \in \{1, \ldots, l\}\}$. This might seem as a discrepancy, but the sample $\mathbf{x}$ can be interpreted as a mixture of Dirac probability measures $p_\mathbf{x} = \frac{1}{l} \sum_{i=1}^l \delta_{x_i}$. By definition of $p_\mathbf{x}$, we immediatelly get

$$\int \phi(x) \mathrm{d}p_\mathbf{x}(x) = \int \frac{1}{l} \sum_{i=1}^l \phi(x) \mathrm{d}\delta_{x_i}(x) = \frac{1}{l} \sum_{i=1}^l \phi(x_i),$$

from which it easy to recover Equation (4). Since $p_\mathbf{x}$ approaches $p$ as $l$ increases, $f(\mathbf{x})$ can be seen as an estimate of $f(p)$. Indeed, if the non-linearities in neural networks implementing functions $\phi$ and $\psi$ are continuous, the function $f$ is bounded and from Hoeffding's inequality (Hoeffding, 1963) it follows that $P(|f(p) - f(\mathbf{x})| \geq t) \leq 2 \exp(-ct^2 l^2)$ for some constant $c > 0$.

## 3   UNIVERSAL APPROXIMATION THEOREM FOR PROBABILITY SPACES

To extend Theorem 1 to spaces of probability measures, the following definition introduces the set of functions which represent the layer that embedds probability measures into $\mathbb{R}$.

**Definition 3.** *For any $\mathcal{X}$ and set of functions $\mathcal{F} \subset \{f : \mathcal{X} \to \mathbb{R}\}$, we define $\mathcal{A}^\mathcal{F}$ as*

$$\mathcal{A}^\mathcal{F} = \left\{ f : \mathcal{P}_\mathcal{X} \to \mathbb{R} \,\middle|\, f(p) = b + \sum_{i=1}^m w_i \int_\mathcal{X} f_i(x) \mathrm{d}p(x), m \in \mathbb{N}, w_i, b \in \mathbb{R}, f_i \in \mathcal{F} \right\}. \tag{6}$$

$\mathcal{A}^\mathcal{F}$ can be viewed as an analogy of affine functions defined by Equation (1) in the context of probability measures $\mathcal{P}_\mathcal{X}$ on $\mathcal{X}$.

*Remark.* Let $\mathcal{X} \subset \mathbb{R}^d$ and suppose that $\mathcal{F}$ only contains the basic projections $\pi_i : x \in \mathbb{R}^d \mapsto x_i \in \mathbb{R}$. If $\mathcal{P}_\mathcal{X} = \{\delta_x | x \in \mathcal{X}\}$ is the set of Dirac measures, then $\mathcal{A}^\mathcal{F}$ coincides with $\mathcal{A}^d$.

Using $\mathcal{A}^\mathcal{F}$, the following definition extends the $\Sigma$-networks from Theorem 1 to probability spaces.

**Definition 4** ($\Sigma$-networks)**.** *For any $\mathcal{X}$, set of functions $\mathcal{F} = \{f : \mathcal{X} \to \mathbb{R}\}$, and a measurable function $\sigma : \mathbb{R} \to \mathbb{R}$, let $\Sigma(\sigma, \mathcal{A}^\mathcal{F})$ be class of functions $f : \mathcal{P}_\mathcal{X} \to \mathbb{R}$*

$$\Sigma(\sigma, \mathcal{A}^\mathcal{F}) = \left\{ f : \mathcal{P}_\mathcal{X} \to \mathbb{R} \,\middle|\, f(p) = \sum_{i=1}^n \alpha_i \sigma(a_i(p)), n \in \mathbb{N}, \alpha_i \in \mathbb{R}, a_i \in \mathcal{A}^\mathcal{F} \right\}. \tag{7}$$

The main theorem of this work can now be presented. As illustrated in a corollary below, when applied to $\mathcal{F} = \Sigma(\sigma, \mathcal{A}^d)$ it states that three-layer neural networks, where first two layers are non-linear interposed with an integration (average) layer, allow arbitrarily precise approximations of continuous function on $\mathcal{P}_\mathcal{X}$. (In other words this class of networks is dense in $C(\mathcal{P}_\mathcal{X}, \mathbb{R})$.)

**Theorem 2.** *Let $\mathcal{P}_\mathcal{X}$ be a compact set of Borel probability measures on a metric space $\mathcal{X}$, $\mathcal{F}$ be a set of continuous functions dense in $C(\mathcal{X}, \mathbb{R})$, and finally $\sigma : \mathbb{R} \to \mathbb{R}$ be a measurable non-polynomial function. Then the set of functions $\Sigma(\sigma, \mathcal{A}^\mathcal{F})$ is dense in $C(\mathcal{P}_\mathcal{X}, \mathbb{R})$.*

Using Lemma 1.1, an immediate corollary is that a similar result holds for measurable funcitons:

**Corollary 1** (Density of MIL NN in $M(\mathcal{P}_\mathcal{X}, \mathbb{R})$)**.** *Under the assumptions of Theorem 2, $\Sigma(\sigma, \mathcal{A}^\mathcal{F})$ is $\rho_\mu$-dense in $M(\mathcal{P}_\mathcal{X}, \mathbb{R})$ for any finite Borel measure $\mu$ on $\mathcal{X}$.*

The proof of Theorem 2 is similar to the proof of Theorem 2.4 from Hornik (1991). One of the ingredients of the proof is the classical Stone-Weierstrass theorem (Stone, 1948). Recall that a collection of functions is an algebra if it is closed under multiplication and linear combinations.

**Stone-Weierstrass Theorem.** *Let $\mathcal{A} \subset C(\mathcal{K}, \mathbb{R})$ be an algebra of functions on a compact $\mathcal{K}$. If*

*(i) $\mathcal{A}$ separates points in $\mathcal{K}$: $(\forall x, y \in K,\ x \neq y)(\exists f \in \mathcal{A}) : f(x) \neq f(y)$ and*

*(ii) $\mathcal{A}$ vanishes at no point of $\mathcal{K}$: $(\forall x \in K)(\exists f \in \mathcal{A}) : f(x) \neq 0$,*

*then the uniform closure of $\mathcal{A}$ is equal to $C(\mathcal{K}, \mathbb{R})$.*

Since $\Sigma(\sigma, \mathcal{A}^{\mathcal{F}})$ is not closed under multiplication, we cannot apply the SW theorem directly. Instead, we firstly prove the density of the class of $\Sigma\Pi$ networks (Theorem 3) which does form an algebra, and then we extend the result to $\Sigma$-networks.

**Theorem 3.** *Let $\mathcal{P}_\mathcal{X}$ be a compact set of Borel probability measures on a metric space $\mathcal{X}$, and $\mathcal{F}$ be a dense subset of $C(\mathcal{X}, \mathbb{R})$. Then the following set of functions is dense in $C(\mathcal{P}_\mathcal{X}, \mathbb{R})$:*

$$\Sigma\Pi(\mathcal{F}) = \left\{ f : \mathcal{P}_\mathcal{X} \to \mathbb{R} \,\middle|\, f(p) = \sum_{i=1}^{n} \alpha_i \prod_{j=1}^{l_i} \int f_{ij}\mathrm{d}p,\ n, l_i \in \mathbb{N}, \alpha_i \in \mathbb{R}, f_{ij} \in \mathcal{F} \right\}.$$

The proof shall use the following immediate corollary of Lemma 9.3.2 from Dudley (2002).

**Lemma 3.1** (Lemma 9.3.2 of Dudley (2002)). *Let $(\mathcal{K}, \rho)$ be a metric space and let $p$ and $q$ be two Borel probability measures on $\mathcal{K}$. If $p \neq q$, then we have $\int f\mathrm{d}p \neq \int f\mathrm{d}q$ for some $f \in C(\mathcal{K}, \mathbb{R})$.*

*Proof of Theorem 3.* Since $\Sigma\Pi(\mathcal{F})$ is clearly an algebra of continuous functions on $\mathcal{P}_\mathcal{X}$, it suffices to verify the assumptions of the SW theorem (separation and non-vanishing properties).

(i) *Separation:* Let $p_1, p_2 \in \mathcal{P}_\mathcal{X}$ be distinct. By Lemma 3.1 there is some $\epsilon > 0$ and $f \in C(\mathcal{X}, \mathbb{R})$ such that $\int f dp_1 - \int f dp_2 = 3\epsilon$. Since $\mathcal{F}$ is dense in $C(\mathcal{X}, \mathbb{R})$, there exists $g \in \mathcal{F}$ such that $\max_{x \in \mathcal{X}} |f(x) - g(x)| < \epsilon$. Using triangle inequality yields

$$\left| \int f\mathrm{d}p_1 - \int f\mathrm{d}p_2 \right| = \left| \int f(x) - g(x) + g(x)\mathrm{d}p_1(x) - \int f(x) - g(x) + g(x)\mathrm{d}p_2(x) \right|$$

$$\leq \left| \int f(x) - g(x)\mathrm{d}p_1(x) \right| + \left| \int f(x) - g(x)\mathrm{d}p_2(x) \right|$$

$$+ \left| \int g(x)\mathrm{d}p_1(x) - \int g(x)\mathrm{d}p_2(x) \right|$$

$$\leq 2\epsilon + \left| \int g\mathrm{d}p_1 - \int g\mathrm{d}p_2 \right|$$

Denoting $f_g(p) = \int g\mathrm{d}p$, it is trivial to see that $f_g \in \Sigma\Pi(\mathcal{F})$. It follows that $\epsilon \leq |f_g(p_1) - f_g(p_2)|$, implying that $\Sigma\Pi(\mathcal{F})$ separates the points of $\mathcal{X}$.

(ii) *Non-vanishing:* Let $p \in \mathcal{P}_\mathcal{X}$. Choose $f \in C(\mathcal{X}, \mathbb{R})$ such that $\int f(x) = 1$. Since $\mathcal{F}$ is dense in $C(\mathcal{X}, \mathbb{R})$ there exists $g \in \mathcal{F}$ such that $\max_{x \in \mathcal{X}} |f(x) - g(x)| \leq \frac{1}{2}$. Since $\int |f - g|\mathrm{d}p \leq \frac{1}{2}$, we get

$$1 = \int f\mathrm{d}p = \int (f - g + g)\,\mathrm{d}p = \int (f(x) - g(x))\mathrm{d}p(x) + \int g\,\mathrm{d}p$$

$$\leq \int |f(x) - g(x)|\mathrm{d}p(x) + \int g\,\mathrm{d}p \leq \frac{1}{2} + \int g\,\mathrm{d}p.$$

Denote $f_g(q) = \int g\,\mathrm{d}q$, $f_g \in \Sigma\Pi(\mathcal{F})$. It follows that $f_g(p) \geq \frac{1}{2}$, and hence $\Sigma\Pi(\mathcal{F})$ vanishes at no point of $\mathcal{P}_\mathcal{X}$.

Since the assumptions of SW theorem are satisfied, $\Sigma\Pi(\mathcal{F})$ is dense in $C(\mathcal{P}_\mathcal{X}, \mathbb{R})$. $\qquad\square$

The following simple lemma will be useful in proving Theorem 2.

**Lemma 3.2.** *If $\mathcal{G}$ is dense in $C(\mathcal{Y}, \mathbb{R})$, then for any $h : \mathcal{X} \to \mathcal{Y}$, the collection of functions $\{g \circ h \,|\, g \in \mathcal{G}\}$ is dense in $\{\phi \circ h \,|\, \phi \in C(\mathcal{Y}, \mathbb{R})\}$.*

*Proof.* Let $g \in C(\mathcal{Y}, \mathbb{R})$ and $g^* \in \mathcal{G}$ be such that $\max_{y \in \mathcal{Y}} |g(y) - g^*(y)| \leq \epsilon$. Then we have

$$\max_{x \in \mathcal{X}} |f(x) - g^*(h(x))| = \max_{x \in \mathcal{X}} |g(h(x)) - g^*(h(x))| \leq \max_{y \in \mathcal{Y}} |g(y) - g^*(y)| \leq \epsilon, \quad (8)$$

which proves the lemma. $\qquad\square$

*Proof of Theorem 2.* Theorem 2 is a consequence of Theorem 3 and $\Sigma$-networks being dense in $C(\mathbb{R}^k, \mathbb{R})$ for any $k$.

Let $\mathcal{X}, \mathcal{F}, \mathcal{P}_\mathcal{X}$, and $\sigma$ be as in the assumptions of the theorem. Let $f^* \in C(\mathcal{P}_\mathcal{X}, \mathbb{R})$ and fix $\epsilon > 0$. Then, there exist $f \in \Sigma\Pi(\mathcal{F})$ such that $\max_{p \in \mathcal{P}_\mathcal{X}} |f(p) - f^*(p)| \leq \frac{\epsilon}{2}$. This function is of the form

$$f(p) = \sum_{i=1}^{n} \alpha_i \prod_{j=1}^{l_i} \int f_{ij} \mathrm{d}p$$

for some $\alpha_i \in \mathbb{R}$ and $f_{ij} \in \mathcal{F}$. Moreover $f$ can be written as a composition $f = g \circ h$, where

$$h : p \in \mathcal{P}_\mathcal{X} \mapsto \left( \int f_{11}\mathrm{d}p, \int f_{12}\mathrm{d}p, \ldots, \int f_{nl_n}\mathrm{d}p \right), \quad (9)$$

$$g : (x_{11}, x_{12}, \ldots, x_{nl_n}) \mapsto \sum_{i=1}^{n} \alpha_i \prod_{j=1}^{l_i} x_{ij} \in \mathbb{R}. \quad (10)$$

Denoting $s = \sum_{i=1}^{n} l_i$, we identify the range of $h$ and the domain of $g$ with $\mathbb{R}^s$.

Since $g$ is clearly continuous and $\Sigma(\sigma, \mathcal{A}^s)$ is dense in $C(\mathbb{R}^s, \mathbb{R})$ (by Theorem 1) there exists $\tilde{g} \in \Sigma(\sigma, \mathcal{A}^s)$ such that $\max_{y \in \mathcal{Y}} |g(y) - \tilde{g}(y)| \leq \frac{\epsilon}{2}$. It follows that $\tilde{f} := \tilde{g} \circ h$ satisfies

$$\begin{aligned}
\max_{p \in \mathcal{P}_\mathcal{X}} |f^*(p) - \tilde{f}(p)| &= \max_{p \in \mathcal{P}_\mathcal{X}} |f^*(p) - f(p) + f(p) - \tilde{g}(h(p))| \\
&\leq \max_{p \in \mathcal{P}_\mathcal{X}} |f^*(p) - f(p)| + \max_{p \in \mathcal{P}_\mathcal{X}} |f(p) - \tilde{g}(h(p))| \\
&\leq \frac{\epsilon}{2} + \frac{\epsilon}{2} = \epsilon && \text{(by Lemma 3.2).}
\end{aligned}$$

Since $\tilde{g} \in \Sigma(\sigma, \mathcal{A}^s)$, it is easy to see that $\tilde{f}$ belongs to $\Sigma(\sigma, \mathcal{A}^\mathcal{F})$, which concludes the proof. $\qquad\square$

The function $h$ in the above construction (Equation (9)) can be seen as a feature extraction layer embedding the space of probability measures into a Euclidean space. It is similar to a mean-map (Smola et al., 2007; Sriperumbudur et al., 2008) — a well-established paradigm in kernel machines — in the sense that it characterizes a class of probability measures but, unlike mean-map, only in parts where positive and negative samples differ.

## 4 UNIVERSAL APPROXIMATION THEOREM FOR PRODUCT SPACES

The next result is the extension of the universal approximation theorem to product spaces, which naturally occur in structured data. The motivation here is for example if one sample consists of some real vector $x$, set of vectors $\{x_i^1\}_{i=1}^{n_1}$ and another set of vectors $\{x_i^2\}_{i=1}^{n_2}$.

**Theorem 4.** *Let $\mathcal{X}_1 \times \cdots \times \mathcal{X}_l$ be a Cartesian product of metric compacts, $\mathcal{F}_i$, $i = 1, \ldots, l$ be dense subsets of $C(\mathcal{X}_i, \mathbb{R})$, and $\sigma : \mathbb{R} \to \mathbb{R}$ be a measurable function which is not an algebraic polynomial. Then $\Sigma(\sigma, \mathcal{A}^{\mathcal{F}_1 \times \ldots \times \mathcal{F}_l})$ is dense in $C(\mathcal{X}_1 \times \cdots \times \mathcal{X}_l, \mathbb{R}), where$*

$$\Sigma(\sigma, \mathcal{A}^{\mathcal{F}_1 \times \ldots \times \mathcal{F}_l}) = \left\{ f : \mathcal{X}_1 \times \cdots \times \mathcal{X}_l \to \mathbb{R} \,\middle|\, f(x_1, \ldots, x_l) = \sum_{i=1}^{n} \alpha_i \sigma \left( b_i + \sum_{j=1}^{l} w_{ij} a_{ij}(x_i) \right), \right.$$

$$\left. n \in \mathbb{N}, \alpha_i, b_i, w_{ij} \in \mathbb{R}, a_{ij}(x) \in \mathcal{F}_j \right\}.$$

The theorem is general in the sense that it covers cases where some $\mathcal{X}_i$ are compact sets of probability measures as defined in Section 2, some are subsets of Euclidean spaces, and others can be general compact spaces for which the corresponding sets of continuous function are dense in $C(\mathcal{X}_i, \mathbb{R})$.

The theorem is a simple consequence of the following corollary of Stone-Weierstrass theorem.

**Corollary 2.** *For $\mathcal{K}_1$ and $\mathcal{K}_2$ compact, the following set of functions is dense in $C(\mathcal{K}_1 \times \mathcal{K}_2, \mathbb{R})$*

$$\left\{ f : \mathcal{K}_1 \times \mathcal{K}_2 \to \mathbb{R} \, \middle| \, f(x,y) = \sum_{i=1}^{n} f_i(x)g_i(y), n \in \mathbb{N}, f_i \in C(\mathcal{K}_1, \mathbb{R}), g_i \in C(\mathcal{K}_2, \mathbb{R}) \right\}.$$

*Proof of Theorem 4.* The proof is technically similar to the proof of Theorem 2. Specifically, let $f$ be a continuous function on $\mathcal{X}_1 \times \cdots \times \mathcal{X}_l$ and $\epsilon > 0$. By the aforementioned corollary of the SW theorem, there are some $f_{ij} \in \mathcal{F}_j, i = 1, \ldots, n, j = 1, \ldots, l$ such that

$$\max_{(x_1, \ldots, x_l) \in \mathcal{X}_1 \times \cdots \times \mathcal{X}_l} |f(x) - \sum_{i=1}^{n} \prod_{j=1}^{l} f_{ij}(x_i)| < \epsilon.$$

Again, the above function can be written as a composition of two functions

$$h : x \in \mathcal{X}_1 \times \cdots \times \mathcal{X}_l \mapsto (f_{11}(x_1), f_{12}(x_2), \ldots, f_{nl}(x_l)) \in \mathbb{R}^{nl}, \tag{11}$$

$$g : x \in \mathbb{R}^{nl} \mapsto \sum_{i=1}^{n} \prod_{j=1}^{l} x_{ij} \in \mathbb{R}. \tag{12}$$

Since $g$ is continuous, Theorem 1 can be applied to obtain a function $\tilde{g}$ of the form $\tilde{g}(x) = \sum_{i=1}^{\tilde{n}} \alpha_i \sigma(b_i + a_i(x))$, for some $\alpha_i \in \mathbb{R}$ and $a_i \in \mathcal{A}^{nl}$, which approximates $g$ with error at most $\epsilon$. Applying Lemma 3.2 to $g$, $h$, and $\tilde{g}$ concludes the proof.

$\square$

## 5   MULTI-INSTANCE LEARNING AND TREE STRUCTURED DATA

The following corollary of Theorem 2 justifies the *embedding paradigm* of Zaheer et al. (2017); Edwards & Storkey (2017); Pevný & Somol (2017) to MIL problems:

**Corollary 3** (Density of MIL NN in $C(\mathcal{P}_\mathcal{X}, \mathbb{R})$). *Let $\mathcal{X}$ be a compact subset of $\mathbb{R}^d$ and $\mathcal{P}_\mathcal{X}$ a compact set of probability measures on $\mathcal{X}$. Then any function $f \in C(\mathcal{P}_\mathcal{X}, \mathbb{R})$ can be arbitrarily closely approximated by a three-layer neural network composed of two non-linear layers with integral (mean) aggregation layer between them, and a linear output layer.*

If $\mathcal{F}$ in Theorem 2 is set to all feed-forward networks with a single non-linear layer (that is, when $\mathcal{F} = \Sigma(\sigma, \mathcal{A}^d)$) then the theorem says that for every $f \in C(\mathcal{P}_\mathcal{X}, \mathbb{R})$ and $\epsilon > 0$, there is some $\tilde{f} \in \Sigma(\sigma, \mathcal{A}^{\Sigma(\sigma, \mathcal{A}^d)}))$ such that $\max_{p \in \mathcal{P}_\mathcal{X}} |f(p) - f^*(p)| < \epsilon$. This $\tilde{f}$ can be written as

$$\tilde{f}(p) = \mathbf{W_1} \left( \sigma \left( \mathbf{W_2} \left( \int \mathbf{W_3} \left( \sigma \left( \mathbf{W_4} x \right) \right) \mathrm{d}p(x) \right) \right) \right),$$

where for brevity the bias vectors are omitted, $\sigma$ and $\int$ are element-wise, and $\mathbf{W}_{(\cdot)}$ are matrices of appropriate sizes. Since the integral in the middle is linear with respect to the matrix-vector multiplication, $\mathbf{W}_2$ and $\mathbf{W}_3$ can be replaced by a single matrix, which proves the corollary:

$$\tilde{f}(p) = \mathbf{W_1} \left( \sigma \left( \mathbf{W_2} \left( \int \sigma \left( \mathbf{W_3} x \right) \mathrm{d}p(x) \right) \right) \right).$$

Since Theorem 2 does not have any special conditions on $\mathcal{X}$ except to be compact metric space and $\mathcal{F}$ to be continuous and uniformly dense in $\mathcal{X}$, the theorem can be used as an induction step and the construction can be repeated.

For example, consider a compact set of probability measures $\mathcal{P}_{\mathcal{P}_\mathcal{X}}$ on a $\mathcal{P}_\mathcal{X}$. Then the space of neural networks with four layers is dense in $C(\mathcal{P}_{\mathcal{P}_\mathcal{X}}, \mathbb{R})$. The network consists of three non-linear layers with integration (mean) layer between them, and the last layer which is linear.

The above induction is summarized in the following theorem.

**Theorem 5.** *Let $\mathcal{S}$ be the class of spaces which (i) contains all compact subsets of $\mathbb{R}^d$, $d \in \mathbb{N}$, (ii) is closed under finite cartesian products, and (iii) for each $\mathcal{X} \in \mathcal{S}$ we have $\mathcal{P}(\mathcal{X}) \in \mathcal{S}$.[1] Then for each $\mathcal{X} \in \mathcal{S}$, every continuous function on $\mathcal{X}$ can be arbitrarilly well approximated by neural networks.*

By Lemma 1.1, an analogous result holds for measurable functions.

*Proof.* It suffices to show that $\mathcal{S}$ is contained in the class $\mathcal{W}$ of all compact metric spaces $\mathcal{X}$ for which functions realized by neural networks are dense in $C(\mathcal{W}, \mathbb{R})$. By Theorem 1, $\mathcal{W}$ satisfies (i). The properties (ii) and (iii) hold for $\mathcal{W}$ by Theorems 4 and 2. It follows that $\mathcal{W} \supset \mathcal{S}$. □

## 6 RELATED WORK

Works most similar to this one are on kernel mean embedding (Smola et al., 2007; Sriperumbudur et al., 2008), showing that a probability measure can be uniquely embedded into high-dimensional space using characteristic kernel. Kernel mean embedding is widely used in Maximum Mean Discrepancy (Gretton et al., 2012) and in Support Measure Machines (Muandet et al., 2012; Christmann & Steinwart, 2010), and is to our knowledge the only algorithm with proven approximation capabilities comparable to the present work. Unfortunately its worst-case complexity of $O(l^3 b^2)$, where $l$ is the number of bags and $b$ is the average size of a bag, prevents it from scaling to problems above thousands of bags.

The MIL problem has been studied in Vinyals et al. (2016) proposing to use a LSTM network augmented by memory. The reduction from sets to vectors is indirect by computing a weighted average over elements in an associative memory. Therefore the aggregation tackled here is an integral part of architecture. The paper lacks any approximation guarantees.

Problems, where input data has a tree structure, naturally occur in language models, where they are typically solved by recurrent neural networks (Irsoy & Cardie, 2014; Socher et al., 2013). The difference between these models is that the tree is typically binary and all leaves are homogeneous in the sense that either each of them is a vector representation of a word or each of them is a vector representation of an internal node. Contrary, here it is assumed that the tree can have an arbitrary number of heterogeneous leaves following a certain fixed scheme.

Due to lack of space, the authors cannot list all works on MIL. The reader is instead invited to look at the excellent overview in Amores (2013) and the works listed in the introductory part of this paper.

## 7 CONCLUSION

This work has been motivated by recently proposed solutions to multi-instance learning Zaheer et al. (2017); Pevný & Somol (2017); Edwards & Storkey (2017) and by mean-map embedding of probability measures Sriperumbudur et al. (2008). It generalizes the universal approximation theorem of neural networks to compact sets of probability measures over compact subsets of Euclidean spaces. Therefore, it can be seen as an adaptation of the mean-map framework to the world of neural networks, which is important for comparing probability measures and for multi-instance learning, and it proves the soundness of the constructions of Pevný & Somol (2017); Edwards & Storkey (2017).

The universal approximation theorem is extended to inputs with a tree schema (structure) which, being the basis of many data exchange formats like JSON, XML, ProtoBuffer, Avro, etc., are nowadays ubiquitous. This theoretically justifies applications of (MIL) neural networks in this setting.

As the presented proof relies on the Stone-Weierstrass theorem, it restricts non-linear functions in neural networks to be continuous in all but the last non-linear layer. Although this does not have an impact on practical applications (all commonly use nonlinear functions within neural networks are continuous) it would be interesting to generalize the result to non-continuous non-linearities, as has been done for feed-forward neural networks in Leshno et al. (1993).

---

[1]Here we assume that $\mathcal{P}(\mathcal{X})$ is endowed with the metric $\rho^*$ from Section 2.

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
