# OpenReview forum: "Approximation capability of neural networks on sets of probability measures and tree-structured data"
_ICLR.cc/2019/Conference_

### Official Review · AnonReviewer2 · 2018-10-29
**The paper proposes a quite straightforward extension of standard results about universal approximation by neural networks on complex domains.**

**Rating:** 4
**Confidence:** 5

**Review:**

The authors study in this paper the approximation capabilities of neural networks for real valued functions on probability measure spaces (and on tree structured domains).

The first step of the paper consists in extending standard NN results to probability measure spaces, that is rather than having finite dimensional vectors as inputs, the NN considered here have probability measures as inputs. The extension to this case is straightforward and closely related to older extension on infinite dimensional spaces (see for instance the seminal paper of Stinchcombe https://doi.org/10.1016/S0893-6080(98)00108-7 and e.g. http://dx.doi.org/10.1016/j.neunet.2004.07.001 for an application to NN with functional inputs). Nothing quite new here.

In addition, and exactly as in the case of functional inputs, the real world neural networks do not implement what is covered by the theorem but only an approximation of it. This is acknowledged by the authors at the end of Section 2 but in a way that is close to hand waving. Indeed while the probability distribution point is valuable and gives interesting tools in the MIL context, the truth is that we have no reason to assume the bag sizes will grow to infinite or even will be under our control. In fact there are many situations were the bag sizes are part of the data (for instance when a text is embedded in a vector space word by word and then represented as a bag of vectors). Thus proving some form of universal approximation in the multiple instance learning context would need to take this fact into account, something that is not done at all here.

Therefore I believe the contribution of this paper to be somewhat limited.

---

> ### Author Response · Authors · 2018-11-19
> **importance**
>
> Without disagreeing with the arguments regarding novelty and bag sizes, we would like to add that for the purposes of MIL NN, being able to work with general probability measures is more general than being able to work with functions in L^p(mu) as in [1], since mu has to be fixed and this only gives measures which are absolutely continuous w.r.t. mu. We also hope that for application to MIL NN, our result should be more accessible than [1] --- while our Theorem 5 gives the approximation property for MIL NN directly, some additional effort is required before being able to apply [1] to specific scenarios (the amount of said effort being quite dependent on the readers background).
>
>
> [1] Rossi, Fabrice, and Brieuc Conan-Guez. "Functional multi-layer perceptron: a non-linear tool for functional data analysis." Neural networks 18.1 (2005): 45-60.

---

### Official Review · AnonReviewer3 · 2018-11-05
**Interesting but...**

**Rating:** 5
**Confidence:** 5

**Review:**

This paper generalizes the universal approximation theorem (usually stated for real functions on some Euclidean space) to real functions on the space of measures (at least a compact set of proba. measures).

This result might be interesting but not really surprising and the paper does not put any new theoretical ideas or proof techniques. The proof is actually almost identical than in the original paper of Hornik, Stinchcombe and White (89) [and not the 91 paper of Hornik as indicated in the paper], the only difference being a trick on the density of f\circ h instead of just considering cos() function.

All in all, the contributions is interesting but really incremental

---

> ### Author Response · Authors · 2018-11-20
> **Key properties should have proofs even if they aren't surprising.**
>
> We do not dispute the novelty of the proof, yet we believe that as the number of applications of AI grows, it becomes important to prove even expected results, as the lack of a proof can help us spot the unsound constructions quicker. The proof itself is important for the field of multi-instance learning, since it has been shown in [1] that the MIL NN architecture it addresses is a considerable improvement over the prior art on a wide range (20) of problems.
>
> [1] Pevný, Tomáš, and Petr Somol. "Using neural network formalism to solve multiple-instance problems." International Symposium on Neural Networks. Springer, Cham, 2017.

---

### Official Review · AnonReviewer1 · 2018-11-05
**Useful result on universality. Probably not extremely relevant to ICLR**

**Rating:** 6
**Confidence:** 3

**Review:**

The paper investigates the approximation properties of a family of neural networks designed to address multi-instance learning (MIL) problems. The authors show that results well-known for standard one layer architectures extend to the MIL models considered. The authors focus on tree-structured domains showing that their analysis applies to these relevant settings.

The paper is well written and easy to follow. In particular the theoretical analysis is clear and pleasant to read.

The main concern is related to the relevance of the result to ICLR. As the authors themselves state, the result is not surprising given the standard universality result of one-layer neural networks (and indeed Thm. 2 heavily relies on this fact to prove the universality of MIL architectures). In this sense the current work might be more suited to a journal venue.

---

> ### Author Response · Authors · 2018-11-13
> **Relevance to learning representation**
>
> The truth is that this work has been inspired by difficulty to use neural networks on security-related problems. As has been written in the introduction, most methods (multi-layer perceptron, convolutional neural networks) assumes samples to have a fixed euclidean dimension, or (recurrent neural networks) being sequences of vectors of a fixed dimension.
>
> But in many domains where you ingesting data using APIs, they comes typically in form of JSON documents (see for example https://www.threatcrowd.org/searchApi/v2/ip/report/?ip=188.40.75.132). This type of data can be elegantly processed by the proposed framework (and the accompanying library). Therefore we believe that it is relevant to ICLR.

---

### Author Response · Authors · 2018-11-12
**Justifying the usefulness**

Dear Reviewers,

we admit that the results aren't "surprising". But taking into account the recent paper [1], we believe the results are important. Ref. [1], published last year at NIPS, studies the same approach as described in our paper (previously independently proposed in [2, 3]), but justifies the construction only for a limited case of probability distributions over finite sets. Our paper fills this gap by extending the justification to probability distributions with infinite support.

The construction seems to be versatile, as it has been recently used in many cited papers, for example in [4] (cited 37 times) it is used within a reasoning module, in [5] (cited 155 times) it is used to learn messages in message passing algorithms for graphs, and in [6] (cited 273 times) it is used for 3D scene recognition.

Taking the above into account, we think that the proof has its place.

[1] Zaheer, Manzil, et al. "Deep sets." Advances in Neural Information Processing Systems. 2017.

[2] Edwards, Harrison, and Amos Storkey. "Towards a neural statistician." arXiv preprint arXiv:1606.02185 (2016).

[3] Pevny, Tomas, and Petr Somol. "Using Neural Network Formalism to Solve Multiple-Instance Problems." arXiv preprint arXiv:1609.07257 (2016).

[4] Santoro, Adam, et al. "A simple neural network module for relational reasoning." Advances in neural information processing systems. 2017

[5] Lin, Guosheng, et al. "Deeply learning the messages in message passing inference." Advances in Neural Information Processing Systems. 2015.

[6] (Qi, Charles R., et al. "Pointnet: Deep learning on point sets for 3d classification and segmentation." Proc. Computer Vision and Pattern Recognition (CVPR), 2017.

---

### Meta-Review · Area_Chair1 · 2018-12-16
**Rewrite needed to address importance of result**

**Confidence:** 3
**Recommendation:** Reject

**Metareview:**

Several reviewers thought the results were not surprising in light of existing universality results, and thought the results were of limited relevance, given that the formalization is not quite in line with real-world networks for MIL. The authors draw out some further justifications in the rebuttal. These should be reintegrated. I agree with the general criticisms regarding relevance to ICLR. Ultimately, this work may belong in a journal.